# Inhibitory effects of black cumin on the formation of heterocyclic aromatic amines in meatball

Emel Oz *

Department of Food Engineering, Faculty of Agriculture, Ataturk University, Erzurum, Turkey

* emel.oz@atauni.edu.tr

## Abstract

The impact of black cumin usage on some qualitative properties and formation of heterocyclic aromatic amines (HAAs) in meatball production was investigated. It was found that black cumin usage rate, cooking process and temperature had a significant effect (p<0,01) on the water content, pH, and thiobarbituric acid reactive substances (TBARS) values of meatballs. On the other hand, black cumin usage significantly (p<0,01) reduced the water content and cooking loss. The water content and cooking loss of the meatballs decreased with increases in the usage rate. While IQx, IQ, MeIQ, 7,8-DiMeIQx, 4,8-DiMeIQx, AαC, and MeAαC could not be detected in meatballs, varying amounts of MeIQx (up to 1,53 ng/g) and PhIP (up to 1,22 ng/g) were determined. The total amounts of HAAs ranged between non-detected (nd) to 2,75 ng/g. Both the usage rate and cooking temperature had a very significant effect (p<0,01) on the total contents of HAAs. The total amounts of HAAs were decreased in correlation with the increases in the usage rate; the proportion which is increased when the cooking temperature increased as well. Results of the present study suggested that addition of black cumin may have a substantial role in decreasing the TBARS value, cooking loss, and HAA contents during meatball production. Therefore, using of black cumin in meatball production has been suggested.

## Introduction

Foods, complex mixture of the micro compounds such as essential trace elements and vitamins and the macro compounds such as protein, fat, and carbohydrates, play an important role in living organisms with other ingredients such as ditary fibers and antioxidants. On the other hand, various hazards, including microbiological and chemicals can be introduced into the food chain from farm to table. Therefore, nutition not only plays a pivotal role in the prevention of some diseases but also may be considered as a risk factor for some other emerging diseases [1].

Some epidemiological investigations have revealed the role of nutrition in the development of many comon cancers [2,3]. Recently, it has been widely accepted that some mutagenic and/ or carcinogenic substances are formed during cooking of foods rich in protein, such as meat

**Funding:** There is not any funding for the present study. The funders had no role in study design, data collection and analysis, decision to publish, or preparation of the manuscript.

**Competing interests:** The author has declared that no competing interests exist.

[4,5]. The group of heterocyclic aromatic amines (HAAs) is of these substances [6]. In 1977, it was reported that some chemical compounds were formed in products such as meat and fish cooked at high temperatures and these compounds were later defined as HAAs, based on their chemical structures [7].

HAAs have two basic chemical groups; 1) Aminoimidazoarenes, IQ type or thermal HAAs, are formed during cooking of food at temperatures of 150–300 ˚C. 2) Aminocarbolines, non-IQ type or pyrolitic HAAs, are formed at temperatures above 300 ˚C [3]. Precursors of HAAs are creatine/creatinine, reducing sugars, amino acids, peptides and proteins, heat and mass transfer, lipids and lipid oxidation [6,8]. Compared to other known food mutagens, HAAs were found to be 100-fold more mutagenic than aflatoxin B1 and 2000-fold more mutagenic than benzo[a]pyrene [3]. On the other hand, according to the International Agency for Research on Cancer (IARC), IQ has been classified as a possible human carcinogen (class 2A) and MelQ, MelQx, PhIP, AαC, MeAαC, Trp-P-1, Trp-P-2, and Glu-P-1 are labelled as probable human carcinogen (class 2B) [9]. Therefore, the presence of HAAs in foods should be prevented and/or reduced.

Since radical reactions had an important role in the formation of HAAs, and it is, therefore, expected that antioxidants can reduce the formation of HAAs in meat and meat products. Indeed, it was reported that antioxidants could inhibit the formation of HAAs and this effect was explained due to the ability of antioxidants to interfere in different stages of the formation reactions of HAAs [10,11]. However, same antioxidants had a prooxidant effect depending on their concentration in food and caused an increase in the formation of HAAs. To reduce the formation of HAAs, synthetic and/or natural antioxidants can be added to meat and meat products. Due to their carcinogenic potential, the use of synthetic antioxidants has been banned from European countries and others, such as Japan and Canada, due to their carcinogenic potential [12,13]. Therefore, during the recent years, food rich in natural antioxidants such as plants or spices have gained much attention [14–18].

Black cumin (*Nigella sativa* L.) is a spice belonging to the family of Ranunculaceae [19]. Black cumin is grown in many parts of the world [20]. It is reported that the seeds of black cumin contain dietary proteins (26,7%), fats (28,5%), and carbohydrates (40,0%) [20]. In the Traditional Middle Eastern medicine, the seeds have been used as a remedy for treatment of multiple diseases (a natural remedy for asthma, hypertension, diabetes, inflammation, cough, bronchitis, headache, eczema, fever, dizziness and influenza) for more than 2000 years [19,21]. Black cumin seed essential oil contains high concentrations of thymoquinone (37,6%) and its related compounds such as thymosl and dithymoquinone and p-cymene (31,4%) with minor amounts of longifolene, carvacrol and thymohydroquinone [19,20].

Black seed showed the most potent radical scavenging better antioxidant activity compared to synthetic antioxidants (BHA and BHT) [19]. In addition, it was documented that black cumin had antimicrobial, anticarcinogen, antioxidant effects and antitumor and antiinflammatory activity [20,21]. It is known that the major component in black cumin is thymoquinone compound and this compound is considered to be a nutritional component [22] and thymoquinone is also considered as potent antioxidant, antiinflammatory, anticarcinogen, antibacterial, antifungal, and antimutagenic agent [19,21].

So far, various studies dealing with the addition of natural and synthetic antioxidants and also food additives to meat and meat products have been published [10, 14–17, 23–27], however, to the best of our knowledge, none of the studies are concerned with the effects of black cumin on the formation of HAAs. Therefore, the purpose of the present study was to investigate the influence of adding of black cumin (0,5 and 1%) in meatball preparation on the formation of HAAs in meatball cooked at different cooking temperatures (150˚C, 200˚C and 250˚C) as well as on the various quality criteria of meatballs.

## Materials and method

### Materials

**Raw materials.** Beef muscle (M. *Gluteus medius*) and intermuscular fat were obtained from a local slaughterhouse (Meat and Milk Institution, Erzurum, Turkey). All subcutaneous fats on meat were removed and used after homogenization. The fat content of meatball dough was adjusted to 15% fat with the intermuscular fat from the same carcass. Black cumin was purchased from a spice store in Erzurum.

**HAA standards.** The HAA standards were purchased from Toronto Research Chemicals (Toronto, Ontario, Canada). 4,7,8-TriMeIQx was used as an internal standard. The stock solutions were prepared according to the method described by Oz and Çakmak [28].

**Preparation of meatballs.** The meatball dough with 15% fat was divided into three portions; one portion was used as control group (without addition of black cumin), and black cumin was added to other portions at two different rates (0,50 and 1%). All groups were shaped into meatballs (7 x 1 cm) after one night storage at 4˚C. No salt or spices were added to meatball dough to avoid any possible interaction.

**Cooking conditions.** Hot plate heated to 150˚C, 200˚C and 250˚C without any fat or oil was used for the cooking of the meatballs. All samples were turned over after half the cooking time (8 min). The surface temperature was measured by a digital thermocouple with a surface probe (Testo 926, Lenzkirch, Germany).

**Water content.** Water contents of the meatballs were determined as weight loss of 10 g homogenized samples following dryness at 102˚C for 24 h [29].

**pH value.** pH values were measured in the samples which was homogenized with distilled water (1:10 w/v) using a pH meter (ATI ORION 420, MA 02129, USA). The pH meter was calibrated using buffer solutions (pH 4.0 and pH 7.0) [29].

**Cooking loss.** Weight of the meatballs was measured before and after cooking [30].

**TBARS value.** Thiobarbituric acid reactive substances (TBARS) values of the samples were determined by the method of Kılıç and Richards [31]. The absorbance of the samples was measured at a wavelength of 532 nm aganist blank. 1, 1, 3, 3- tetraethoxypropane was used fort he calculation of k value. Results were expressed as mg malondialdehyde (MDA)/kg sample.

**Determination of HAA content.** The HAAs contnet of the samples was determined according to Messner and Murkovic [32] with minor modifications [1]. Solid phase extraction method (Oasis cartridges, 3 $cm^3$/60 mg Waters, Milford, MA) was used for the analysis. For this aim, 1 g meatball sample was dissolved in 12 ml 1 M NaOH. The suspension was homogenised by using a magnetic stirring for 1 h at 500 rpm at room temperature. The alkaline solution was mixed with 13 g diatomaceous earth (Extrelut NT packaging material, Merck, Darmstadt, Germany) and then poured into empty Extrelut columns. The extractions were made by using 75 ml ethyl acetate and the eluate was passed through coupled Oasis MCX cartridges. The cartridge was washed with 2 ml of 0.1 M HCl and 2 ml MeOH. The analytes were eluted with 2 ml of MeOH-concentrated (25%) ammonia (19/1, v/v). The eluted mixtures were evaporated to dryness at 50˚C and the final extracts were dissolved in 100 μl MeOH just before measurement. A mixture of purified water/acetonitrile/methanol/glacial acetic acid (76/14/8/2, v/v/v/v) at pH 5,0 (adjusted with ammonium hydroxide 25%) was used as Solvent A, while solvent B was acetonitrile (100%). Separation process was conducted on AcclaimTM 120 C18 3 μm (4,6 x 150 mm) Tosoh Bioscience GmbH (Stuttgart, Germany) at 35˚C. The flow rate was 0,7 ml/min. The gradient programme was as follows: 0% B, 0–10 min; 0–23% B, 11–20 min; 23% B, 21–30 min; 0% B, 31–45 min. The injection volume was 10 ml.

**Statistical analyses.** The experiment was a completely random block design with two replicates. The data obtained in the current study were subjected to analysis of variance. The differences between means were evaluated by Duncan's multiple range test.

## Results and discussion

### Analyses of raw materials

Water content, pH, and TBARS values of the meat, intermuscular fat and raw meatballs with 15% fat were shown in Table 1. The obtained results are in line with those reported by others [33–37].

### Water contents of the meatballs

The water contents of the meatballs were given in Table 2. There was a very significant effect (p<0,01) of usage rate of black cumin, cooking process and cooking temperature on the water contents of the meatballs. The addition of black cumin to meatball dough significantly reduced the water content compared to the control group samples (p<0,05). As the usage rate of black cumin increased, the water content of the samples decreased. The decrease in the water content of meatballs with adding the black cumin is attributable to the high dry matter content of the black cumin used in the present study (95,70%). Cooking, as expected, caused a decrease in the water content of the samples; the finding which is consistent with others [28,38]. In addition, it was also determined that the water content of the samples decreased as the cooking temperature increased. del Pulgar et al. [39] reported the three main reasons for water loss in meat during cooking process. First, as temperature increases so does the process of evaporation. Second, heating caused denaturation of myosin and shrinkage of myofibrils, which in turn lead to a reduction in the myofibril's ability to hold water. Finally, a contraction of the perimysial connective tissue causes a compression of the muscle fiber bundles; in accordance this could enhance water release from the meat cut.

### pH values of the meatballs

pH values of the meatballs were also given in Table 2. There was a very significant effect (p<0,01) of usage rate of black cumin, cooking process and cooking temperature on the pH values of the meatballs. Black cumin usage in the meatball preparation significantly increased the pH values compared to the control group samples (p<0,05). The pH of the samples was increased with the increased usage rate of black cumin. The increase in pH value of meatballs with adding the black cumin could be attributed to the high pH value of the black cumin used in the present study (6,22). Cooking, as expected, caused an increase in pH values of the samples and this finding is in accordance with that reported by others [18,37,40]. In addition, in

**Table 1. Water content, pH and TBARS values of the raw materials (mean ± SD).**

|  | n | Water (%) | pH | TBARS (mg MDA/kg) |
|---|---|---|---|---|
| Meat | 2 | 71,33±0,52 a | 5,64±0,09 b | 0,804±0,107 a |
| Intermuscular fat | 2 | 13,97±0,86 c | 6,54±0,29 a | 0,290±0,082 b |
| Meatball | 2 | 64,18±0,77 b | 5,70±0,16 b | 0,656±0,097 a |
| Sign. |  | ** | ** | ** |

**p< 0,01,

SD: Standard Deviation,

Different letters (a-c) in the same column denote significant differences (p<0,05)

**Table 2. The average water content, pH and TBARS values, cooking loss and total HAA contents of the samples (mean ± SD).**

| | n[x] | Water (%) | pH | TBARS (mg MDA/kg) | Cooking Loss (%) | Total HAA (ng/g) |
|---|---|---|---|---|---|---|
| *Usage rate (UR, %)* | | | | | | |
| 0 | 12 | 61,72 ± 3,95a | 5,98 ± 0,14c | 1,228 ± 0,425a | 31,56 ± 4,74a | 1,25 ± 1,22a |
| 0,5 | 12 | 60,78 ± 4,84b | 6,02 ± 0,14b | 1,088 ± 0,283b | 30,02 ± 4,98b | 0,81 ± 0,79b |
| 1 | 12 | 59,66 ± 5,33c | 6,03 ± 0,12a | 1,023 ± 0,234c | 29,07 ± 4,51c | 0,65 ± 0,62c |
| Sign. | | ** | ** | ** | ** | ** |
| *Cooking Process (CP)* | | | | | | |
| Raw | 18 | 64,95 ± 0,38a | 5,89 ± 0,03b | 0,876 ± 0,032b | | |
| Cooked | 18 | 56,49 ± 2,67b | 6,13 ± 0,06a | 1,349 ± 0,316a | | |
| Sign | | ** | ** | ** | | |
| *Cooking Temperature (CT, ℃)* | | | | | | |
| 150 | 12 | 62,25 ± 2,96a | 5,97 ± 0,09c | 0,959 ± 0,100c | 24,38 ± 1,19c | nd |
| 200 | 12 | 60,13 ± 5,16b | 6,02 ± 0,14b | 1,102 ± 0,259b | 31,65 ± 1,25b | 0,58 ± 0,26b |
| 250 | 12 | 59,80 ± 5,53c | 6,04 ± 0,15a | 1,277 ± 0,455a | 34,62 ± 1,30a | 2,03 ± 0,58a |
| Sign. | | ** | ** | ** | ** | ** |
| *Interactions* | | | | | | |
| UR x CP | | ** | ** | ** | | |
| UR x CT | | ns | * | ** | ns | ** |
| CP x CT | | ** | ** | ** | | |
| UR x CP x CT | | ns | * | ** | | |

** $p < 0,01$,

*$p < 0,05$,

ns: Not Significant ($p > 0,05$),

[x]n = 6 for cooking loss values

SD: Standard Deviation, nd: Not detected,

Different letters (a-c) in the same column denote significant differences ($p < 0,05$)

the present study, it was also determined that the pH values of the samples increased as the cooking temperature increased. Oz et al. [40] found pH value of 6,09 and 6,17 in raw and cooked meatballs, respectively. The reason behind the increase in pH value of meat owing to cooking might be attributed to the cleavage of bonds involving imidazole, sulfhydryl and hydroxyl groups as reported by Girard [41].

## TBARS values of the meatballs

TBARS values of the meatballs were also given in Table 2. There was a very significant effect ($p < 0,01$) of usage rate of black cumin, cooking process and cooking temperature on the TBARS values of the meatballs. Adding of black cumin usage in the meatball preparation significantly reduced TBARS values compared to the control group samples ($p < 0,05$). As the usage rate of black cumin increased, the TBARS values of the samples decreased. The antioxidant effect of black cumin on TBARS value was believed to be due to its phenolic compound. It is known that antioxidant activity of phenolic compounds is linked to their high redox potentials that would allow them to act as reducing agents, hydrogen donors [19]. In addition, it was determined that cooking caused an increase in TBARS values of the samples and the cooking temperature increased, TBARS values of the samples increased. Due to the fact that cooking destroys the cellular structure and inactivates enzymes enabling oxygen to become

free from oxymyoglobin, cooking is one of the main reasons of lipid oxidation in meat and meat products [42].

The obtained TBARS value herein with remained below the acceptance threshold of 2 mg MDA/kg [43,44].

## Cooking loss values of the cooked meatballs

Cooking loss values of the meatballs were also given in Table 2. There was a very significant effect (p<0,01) of usage rate of black cumin and cooking temperature on cooking loss values of meatballs. Black cumin usage in meatball preparation significantly reduced cooking loss values compared to control group (p<0,05). As the usage rate of black cumin increased, the cooking loss values of the samples decreased. This could be attributed to the dry matter content (95,70%) of the black cumin used in the current study. It was found that cooking loss values of the samples increased as the cooking temperature increased.

Rodriguez-Estrada et al. [45] reported that there is a decrease in the weights of meat products as a result of the cooking process; it is assumed that water is removed. Cooking not only removes meat juice from the meat but also some water-soluble compounds therein. Indeed, according to Gerber et al. [46], Laroche declared that meat juices removed during cooking contain some compounds such as myofibrillar or sarcoplasmic proteins, collagen, lipids, salt, polyphosphates, and aroma compounds.

## Limit of detection and limit of quantification values and recoveries of HAAs

Limit of detection (LOD) and limit of quantification (LOQ) for HAAs were calculated based on the signal to- noise ratios of 3 and 10, respectively. Recovery rates for the nine HAAs in the samples were determined by the standard addition method. The recovery values of the nine HAAs were ranged between 28,94 and 82,15%. The LOD values were in the range of 0.004 and 0.025 ng/g, while the LOQ values were in between 0,013 and 0,085 ng/g. These values were comparable to those reported in the literature [26,32].

## HAA content of the meatballs

HAA content of the meatballs were shown in Table 3. While IQx, IQ, MeIQ, 7,8-DiMeIQx, 4,8-DiMeIQx, AαC and MeAαC compounds were not detected in any of the analyzed samples, varying levels of MeIQx (up to 1,53 ng/g) and PhIP (up to 1,22 ng/g) were determined.

**Table 3. The HAA content of the samples (ng/g).**

| Cooking Temperature (°C) | Usage Rate (%) | IQx | IQ | MeIQx | MeIQ | 7,8-DiMeIQx | 4,8-DiMeIQx | PhIP | AαC | MeAαC | Total HAA |
|---|---|---|---|---|---|---|---|---|---|---|---|
| 150 | 0 | nd | nd | nd | nd | nd | nd | nd | nd | nd | nd |
|  | 0,5 | nd | nd | nd | nd | nd | nd | nd | nd | nd | nd |
|  | 1 | nd | nd | nd | nd | nd | nd | nd | nd | nd | nd |
| 200 | 0 | nd | nd | 0,90 | nd | nd | nd | nq | nd | nd | 0,90 |
|  | 0,5 | nd | nd | 0,50 | nd | nd | nd | nd | nd | nd | 0,50 |
|  | 1 | nd | nd | 0,35 | nd | nd | nd | nd | nd | nd | 0,35 |
| 250 | 0 | nd | nd | 1,53 | nd | nd | nd | 1,22 | nd | nd | 2,75 |
|  | 0,5 | nd | nd | 1,00 | nd | nd | nd | 0,85 | nd | nd | 1,85 |
|  | 1 | nd | nd | 0,86 | nd | nd | nd | 0,65 | nd | nd | 1,51 |

nd: Not detected

The amount of IQx was below the LOD; the finding which is consistent with others [28,35,47–52]. On the other hand, IQx was determined up to 0,13 ng/g in beef barbecued at up to 300˚C for up to 20 min by Turesky et al. [53], up to 0,39 ng/g in beef fried at up to 300˚C for up to 24 min by Turesky et al. [53], up to 0,61 ng/g in beef fried at 200˚C for up to 6 min by Oz et al. [54], as 1,5 ng/g in grilled beef by Fay et al. [55] and up to 3,65 ng/g in barbecued beef for up to 6 min by Oz et al. [54].

The amount of IQ was below the LOD; the finding which is consistent with others [28,35,47,48]. On the other hand, IQ was determined as 0,02 ng/g in beef fried at 200 ˚C for 12 min by Felton et al. [56], as 0,5 ng/g in grilled beef by Yamaizumi et al. [57], as 7 ng/g in barbe-cued beef at up to 500˚C for 15 min by Rivera et al. [58] and as 10,2 ng/g in beef fried at 180˚C for 20 min by Murkovic et al. [26].

MeIQx was determined in all of the meatballs cooked at 200˚C and 250˚C, however, the compound could not be detected in the meatballs cooked at 150˚C. As the cooking tempera-ture increased, MeIQx content of the samples increased. In addition, as the usage rate of black cumin increased, MeIQx content of the samples decreased. Although MeIQx was not detected in various meat and meat products [18,35,59–63], MeIQx was determined as 0,64 ng/g in fried beef by Wakabayashi et al. [64], as 1 ng/g in beef fried at 250˚C for 12 min by Felton et al. [56], as 2,11 ng/g in broiled beef by Wakabayashi et al. [64], up to 8,3 ng/g in beef fried at 190˚C for up to 13 min by Gross [65] and as 16,4 ng/g in beef fried at 277˚C for 12 min by Thiébaud et al. [66]. The maximum MeIQx content was recorded in cooked beef at a concentration level of 80 ng/g [3].

The amount of MeIQ was below the LOD; the finding which is consistent with others [35,36,40,48,60]. On the other hand, MeIQ was determined up to 0,38 ng/g in beef barbecued at up to 240˚C for up to 12 min by Abdulkarim and Smith [59], as 2,46 ng/g in beef fried at 180˚C for 20 min by Murkovic et al. [26], and as 8 ng/g in barbecued beef at up to 500˚C for 15 min by Rivera et al. [58].

The amount of 7,8-DiMeIQx was below the LOD; the finding which is consistent with oth-ers [50,52,62]. On the other hand, 7,8-DiMeIQx was determined as 0,2 ng/g in grilled beef by Fay et al. [55], and up to 1,75 ng/g in beef fried at up to 275˚C for up to 15 min by Klassen et al. [67].

The amount of 4,8-DiMeIQx was below the LOD; the finding which is consistent with oth-ers [60,62]. On the other hand, 4,8-DiMeIQx was determined up to 1,2 ng/g in beef fried at up to 250˚C for 12 min by Felton et al. [61] and as 4,5 ng/g in beef fried at 277˚C for 12 min by Thiébaud et al. [66]. The maximum 4,8-DiMeIQx content was recorded in cooked beef at a concentration level of 15 ng/g [3].

PhIP was not detected in all of the meatballs cooked at 150˚C and in the meatballs with black cumin and cooked at 200˚C. In the control group meatballs cooked at 200˚C, the com-pound was detected but its amount could not be determined (not quantified). On the other hand, PhIP was determined in all of the meatballs cooked at 250˚C. In the control group meat-balls, as the cooking temperature increased, PhIP content increased. The use of black cumin in the preparation of meatball caused a reduction in PhIP content of the samples cooked at 200˚C and 250˚C. There are studies in the literature showing that PhIP was not detected in various meat and meat products [25,35,36,40,62]. On the other hand, PhIP was determined as 0,56 ng/g in fried beef by Wakabayashi et al. [64], up to 13,3 ng/g in beef fried at up to 250˚C for 12 min by Felton et al. [61] and as 15,7 ng/g in broiled beef by Wakabayashi et al. [64]. The maximum PhIP content was recorded in cooked beef at a concentration level of 182 ng/g [3].

The amount of AαC was below the LOD; the finding which is consistent with others [47,68,69]. On the other hand, AαC was determined as 1,20 ng/g in broiled beef by Wakabaya-shi et al. [64], up to 3,32 ng/g in beef fried at up to 300˚C for up to 24 min by Turesky et al.

[53], up to 7,75 ng/g in beef barbecued at up to 300˚C for 20 min by Turesky et al. [53] and as 21 ng/g in beef fried at 277˚C for 12 min by Thiébaud et al. [66].

The amount of MeAαC was below the LOD; the finding which is consistent with others [36,47,52,68,69]. On the other hand, MeAαC was determined up to 0,14 ng/g in beef fried at up to 300˚C for up to 24 min by Turesky et al. [53] and up to 0,29 ng/g in beef barbecued at up to 300˚C for 20 min by Turesky et al. [53].

Total HAA content of the meatballs ranged from 0,5–2,75 ng/g, whereas in the case of meatballs cooked at 150˚C HAAs were not detected. It was determined that increases in the cooking temperature would increase the total HAAs of both control and meatballs supplemented with black cumin. While the total HAA content of the control group meatballs cooked at 150˚C was not detectable, the total HAA contents of the control group meatballs cooked at 200˚C and 250˚C were 0,90 ng/g and 2,75 ng/g, respectively. On the other hand, the total HAA content of the meatballs including black cumin at 0,5% and cooked at 150˚C was not detectable, whereas the total HAA contents of those meatballs cooked at 200˚C and 250˚C were 0,50 ng/g and 1,85 ng/g, respectively. While the total HAA content of the meatballs including black cumin at 1% and cooked at 150˚C was not detectable, the total HAA contents of those meatballs cooked at 200˚C and 250˚C were 0,35 ng/g and 1,51 ng/g, respectively. It was also found that the use of black cumin in the preparation of meatball caused a reduction in total HAA content of the samples and as the use rate of black cumin increased, the total HAA content of the meatballs decreased. While the total HAA content of the meatballs including black cumin cooked at 150˚C was not detectable, the use of black cumin at 0,5% in meatball production caused a reduction on the total HAA content of the meatballs cooked at 200˚C (44,44%) and at 250˚C (32,73%). Similarly, the use of black cumin at 1% in meatball production also caused a reduction on the total HAA content of the meatballs cooked at 200˚C (32,73%) and at 250˚C (45,09%). On the other hand, the total HAA contents of the meatballs cooked at 200˚C were belonging to MeIQx compound. While the total HAA contents of the control group meatballs cooked at 250˚C were determined to be consisted of MeIQx (55,64%) and PhIP (44,36%), the total HAA contents of the meatballs containing different amounts of black cumin and cooked at the same temperature were consisted of MeIQx (54,05–56,95%) and PhIP (43,05–45,95%).

Oz and Kaya [16] investigated the effect of black pepper on the formation of HAA and found that the total HAA (IQ, MeIQ, MeIQx, 4,8-DiMeIQx, and PhIP) content ranged between 1,40–37,81 ng/g in their control group meatballs cooked at 175–250˚C. The researchers declared that use of black pepper at 1% inhibited total HAA content up to 100%.

Shin et al. [14] found that the addition of 20 g of minced garlic cloves to ground beef patties would reduce the total HAAs (MeIQx, DiMeIQx and PhIP) by 68%. In another study conducted the same group, it was reported that 24.6 ng/g total HAA (MeIQx, DiMeIQx, and PhIP) was determined in control group fried at 225˚C for 20 min [15]. The authors also found that the addition of diallyl disulfide and dipropyl disulfide to ground beef patties inhibited total HAA formation by up to 78% and 70%, respectively.

The use of spices with antioxidant properties in meat preparation could interfere with different stages of HAA formation. The exact mechanism of inhibiting the HAA formation by antioxidants is still not fully understood. However, it has been generally accepted that antioxidants inactivate free radicals. Due to their antiradical activity, antioxidants could act as inhibitors in the mutagens formation [10]. On the other hand, it was justified that prooxidant or antioxidant effect of antioxidants was highly dependent upon the concentration [25]. In this context, Oz et al. [38] investigated the effect of direct addition of conjugated linoleic acid (CLA) to beef chops cooked at 150˚C, 200˚C and 250˚C on formation of HAAs and found that

while direct addition of 0.05% CLA to beef chops increased (3.85–68.75%) total HAA amount at all cooking temperatures, direct addition of 0.1% CLA to the beef chops decreased (18,75–31,54%) the total HAA amount at all cooking temperatures. Similarly, Oz and Çakmak [28] investigated the effects of CLA usage in meatball production on the formation of HAA in meatballs and found that the use of CLA displayed an inhibitory as well as a stimulatory on total HAA content of meatballs depending on the usage rate and cooking temperatures.

In terms of total HAAs content, it is somehow difficult to compare the results of the present study with the others. This is due to the fact that the current study is the first study on the effect of black cumin usage in the preparation of meatball on the formation of HAAs. On the other hand, the amount of total HAAs in the present study was lower than other studies. The little differences might be arise from the type of the meat used, animal feeding conditions, the various cooking conditions, extraction methods, and type of chromatographic detection systems. On the other hand, according to Knize et al. [70], who cited data by Pearson and others, the total amounts of IQ, MeIQx, and DiMeIQx were 7300 ng/g in beef fried at 215°C; the value which is considerably higher than both results of the present study and others [14–16, 28, 38].

In the present study, it is seen that even if 100 g of the control group meatballs at 250°C whose total amount of HAA content is the highest, is eaten, the intake amount is quite far below the maximum acceptable daily consumption (15 μg/day) stated by Skog [5].

## Conclusion

As a conclusion, the results showed that adding black cumin to meatball would reduce the levels of both individual HAAs (MeIQx and PhIP) and total HAAs. The inhibitory effect of black cumin on HAAs could be attributed to the antioxidant effect of its phenolic compounds.

## Acknowledgments

The author is grateful to Professor Fatih Oz for supporting in the analysis of heterocyclic aromatic amines.

## Author Contributions

**Formal analysis:** Emel Oz.

**Investigation:** Emel Oz.

**Methodology:** Emel Oz.

**Supervision:** Emel Oz.

**Validation:** Emel Oz.

**Writing – original draft:** Emel Oz.

**Writing – review & editing:** Emel Oz.

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
