## [Decision Letter · Decision Letter 0]

31 Jul 2019

PONE-D-19-18609

The effect of the use of black cumin in meatball preparation on the formation of heterocyclic aromatic amines

PLOS ONE

Dear Dr. Oz,

Thank you for submitting your manuscript to PLOS ONE. After careful consideration, we feel that it has merit but does not fully meet PLOS ONE’s publication criteria as it currently stands. Therefore, we invite you to submit a revised version of the manuscript that addresses the points raised during the review process.

Please revise this paper.

We would appreciate receiving your revised manuscript by Sep 14 2019 11:59PM. To enhance the reproducibility of your results, we recommend that if applicable you deposit your laboratory protocols in protocols.io, where a protocol can be assigned its own identifier (DOI) such that it can be cited independently in the future. For instructions see: http://journals.plos.org/plosone/s/submission-guidelines#loc-laboratory-protocols

We look forward to receiving your revised manuscript.

Kind regards,

Aneta Agnieszka Koronowicz, PhD

Academic Editor

PLOS ONE

Journal Requirements:

Reviewers' comments:

Reviewer's Responses to Questions

**Comments to the Author**

1. Is the manuscript technically sound, and do the data support the conclusions?

Reviewer #1: Yes

Reviewer #2: Yes

Reviewer #3: Yes

2. Has the statistical analysis been performed appropriately and rigorously? 

Reviewer #1: Yes

Reviewer #2: Yes

Reviewer #3: Yes

3. Have the authors made all data underlying the findings in their manuscript fully available?

Reviewer #1: Yes

Reviewer #2: Yes

Reviewer #3: Yes

4. Is the manuscript presented in an intelligible fashion and written in standard English?

Reviewer #1: Yes

Reviewer #2: Yes

Reviewer #3: No

5. Review Comments to the Author

Reviewer #1: Dear Editor,

The article is deal with the inhibitory effect of black seed on the formation of heterocyclic amines in meatball. The topic is interest and the article is good and easy to follow. It can be published in your journal after done necessary corrections. My specific comments and questions about the article are below.

- The author should use black cumin or Nigella sativa throughout the article.

- Title: The title of the article should be “Inhibitory effects of black cumin on the formation of heterocyclic aromatic amines in meatball”.

- Page 2, line 31: significantly reduced? Give P values in statistical analysis.

- Page 4, line 98-99: No need to this sentence, repetition for the above stated sentence (line 83-84).

- Page 6, line 143: blind or blank?

- Page 7, line 151: distilled water or purified water?

- Page 15, line 349-350: What do you mean? This sentence is not clear.

- Page 16, line 377: Give the references.

Reviewer #2: The paper is very interesting. From chemical point of view, there is nothing new, just another scenario of Maillard reaction. The data are brief and clear. The paper is well written. And it includes the relevant literature. It seems from the data presented that the black cumin can prevent the formation of genotoxic mutagenic amines to some extent. Taking into account nutritional aspects, it is a valuable finding. Therefore, I recommend the present study to be published as it is.

Reviewer #3: In my opinion this manuscript may be published in PLOS ONE only after a major revision. Generally, English must be improved. In this study, the Authors estimated the effect of use of black cumin in meatball preparation on formation of HAAs. In my opinion, it would improve a paper a lot if the Authors added (e.g., in introduction) any information about black cumin as a source of antioxidants. Beside essentiale oil containing mainly carvone and limonen, black cumin seeds may contain also phenolic acids (e.g., caffeic acid), flavonoids, leuco-anthocyanins that may have a crucial role in scavenging of free radicals. The last ones may be responsible for formation of HHAs.

The remaining comments to the Authors are given below.

Page 3 line 58 „aminoimidazoaazoarenes” shouild be changed into „aminoimidazoarenes”

Page 3 line 63 It is not clear if antioxidants are precursors of HAAs ort hey may play a role of inhibitors of HAAs or any others contaminants

Page 3 line 64 „ HAAs were found to be” should be changed as „HAAs content (or level) were found to be 100-fold higher”

Page 3 line 75 According to the Authors „some antioxidants had a prooxidant effect”, however,

the same antioxidants could possess prooxidant or antioxidant activity depending on their concentration in food”

Page 4 line 80 „natural antioxidants” should be changed as „food rich in natural antioxidants such as”, line 83 „protein, fat and carbohydrate” should be changed into „proteins, fats and harbohydrates”

Page 4 line 90 „carvacrola” should be changed into „carvacrol”, line 95 „ antiinflamatory” should be changed as „antiinflamatory activity (or properties)”

Page 4 lines 96&98 „thymokinone” should be changed as „thymoquinone”

Page 4 lines 98&99 The last sentence is the exact repetition what was written before (lines 83&84)

Page 5 line 101 „food ingredients” should be changed as „food additives”

Page 6 line 150 It is not clear in which way the mentioned solvents were used during the solid phase extraction methods, so a more detailed description of the SPE extraction is needed

Page 7 line 153 Please, give a information that the gradient method was used during chromatographic analysis, moreover, 3 mm should be changed into 3µm (in case of particles diameter)

Page 7 line 161 Please, use the same designation in the description of Table 1 as in Table 1 (eg. „meat” in Table 1 and „beef muscle” in line 161

Page 8 line 199 „groups by Girard” should be changed as „ groups as reported by Girard”

Page 13 line 314 „ranged between nd – 2,75 ng/g” should be changed into „ ranged from 0,5 – 2,75 ng/g, whereas in the case of meatballs cooked at 150°C HAAs were not detected”

Page 15 line 355 „radical quenchers and free radical scavenging activity are designation of the same phenomenon – antiradical activity

Page 16 line 373 „the chromatographic detector” should be changed as „ type of chromatographic detection”

Page 16 lines 382&385 The last sentence is not so adequate to the conclusion of this manuscript

Table 2 The Authors have given the values of determinated parameters in first line named as Usage rate (0-1) but nothin is known about temperature of cooking (in case of first line), so detailed information could be helpful for the readers (under the table or in the text),

With respect to the values of parameters given in line No 3 named as cooking temperature, did the Authors give the average values for various usage rate (0–1 %), please to explain

6. PLOS authors have the option to publish the peer review history of their article (what does this mean?). If published, this will include your full peer review and any attached files.

Reviewer #1: No

Reviewer #2: No

Reviewer #3: No

---

## [Author Response · Author response to Decision Letter 0]

1 Aug 2019

Plos One

PONE-D-19-18609

“The effect of the use of black cumin in meatball preparation on the formation of heterocyclic aromatic amines”

Dear Editor,

Thank you for your useful comments. I have modified the manuscript accordingly. The corrections according to reviewers comment were given in yellow in the text. The detailed corrections are also listed below point by point:

Response to the reviewers:

Review Comments to the Author

Reviewer #1: 

Dear Editor,

The article is deal with the inhibitory effect of black seed on the formation of heterocyclic amines in meatball. The topic is interest and the article is good and easy to follow. It can be published in your journal after done necessary corrections. My specific comments and questions about the article are below.

- The author should use black cumin or Nigella sativa throughout the article.

“Black cumin” was used throughout the article. 

- Title: The title of the article should be “Inhibitory effects of black cumin on the formation of heterocyclic aromatic amines in meatball”.

I would like to thank for the reviewer comment. The title of the article has been changed as “Inhibitory effects of black cumin on the formation of heterocyclic aromatic amines in meatball”.

- Page 2, line 31: significantly reduced? Give P values in statistical analysis.

P value has been added to the sentence.

- Page 4, line 98-99: No need to this sentence, repetition for the above stated sentence (line 83-84).

Thank you for the reviewer’s attention. The second sentence has been discarded form the text.

- Page 6, line 143: blind or blank?

“Blind” has been changed as “blank”.

- Page 7, line 151: distilled water or purified water?

“Water” has been changed as “purified water”

- Page 15, line 349-350: What do you mean? This sentence is not clear.

Thank you. The sentence has been modified.

- Page 16, line 377: Give the references.

The references have been added to the text.

Reviewer #2: The paper is very interesting. From chemical point of view, there is nothing new, just another scenario of Maillard reaction. The data are brief and clear. The paper is well written. And it includes the relevant literature. It seems from the data presented that the black cumin can prevent the formation of genotoxic mutagenic amines to some extent. Taking into account nutritional aspects, it is a valuable finding. Therefore, I recommend the present study to be published as it is.

I would like to thank for the reviewer comment.

Reviewer #3: In my opinion this manuscript may be published in PLOS ONE only after a major revision. Generally, English must be improved. In this study, the Authors estimated the effect of use of black cumin in meatball preparation on formation of HAAs. In my opinion, it would improve a paper a lot if the Authors added (e.g., in introduction) any information about black cumin as a source of antioxidants. Beside essentiale oil containing mainly carvone and limonen, black cumin seeds may contain also phenolic acids (e.g., caffeic acid), flavonoids, leuco-anthocyanins that may have a crucial role in scavenging of free radicals. The last ones may be responsible for formation of HHAs.

I would like to thank you for the reviewer’s useful comment. A native speaker has checked the article and the necessary corrections have been done on it. About the antioxidant activity of the black cumin, black cumin seeds contain fixed oil and volatile oil including thymoquinone and monoterpenes such as r-cymene and a-piene. However, it is reported that thymoquinone is major component of black cumin and it is considered as potent antioxidant, anticarcinogenic and antimutagenic. In the present study, usage of black cumin in meatball production inhibited the formation of HAAs (not promoted) and its inhibitory effect could be attributed to its thymoquinone content. The sentence about the antioxidative properties of black cumin has been modified. 

The remaining comments to the Authors are given below.

Page 3 line 58 „aminoimidazoaazoarenes” shouild be changed into „aminoimidazoarenes”

Thank you for the reviewer’s attention. “aminoimidazoaazoarenes” has been changed as “aminoimidazoarenes”.

Page 3 line 63 It is not clear if antioxidants are precursors of HAAs ort hey may play a role of inhibitors of HAAs or any others contaminants

The reviewer is right. Therefore, the word of “antioxidants” has been discarded from the text.

Page 3 line 64 „ HAAs were found to be” should be changed as „HAAs content (or level) were found to be 100-fold higher”

In here, I compared the mutagenicities of HAAs and the other known food mutagens such as aflatoxin B1 and benzo(a)pyrene. However, the sentence has been modified.

Page 3 line 75 According to the Authors „some antioxidants had a prooxidant effect”, however,

the same antioxidants could possess prooxidant or antioxidant activity depending on their concentration in food”

The sentence has been modified.

Page 4 line 80 „natural antioxidants” should be changed as „food rich in natural antioxidants such as”, line 83 „protein, fat and carbohydrate” should be changed into „proteins, fats and harbohydrates”

The corrections have been done. Thank you.

Page 4 line 90 „carvacrola” should be changed into „carvacrol”, line 95 „ antiinflamatory” should be changed as „antiinflamatory activity (or properties)”

Page 4 lines 96&98 „thymokinone” should be changed as „thymoquinone”

The corrections have been done. Thank you.

Page 4 lines 98&99 The last sentence is the exact repetition what was written before (lines 83&84)

Thank you for the reviewer’s attention. The sentence has been discarded form the text.

Page 5 line 101 „food ingredients” should be changed as „food additives”

“food ingredients” has been changed as “food additives”.

Page 6 line 150 It is not clear in which way the mentioned solvents were used during the solid phase extraction methods, so a more detailed description of the SPE extraction is needed

Thank you. The part of SPE has been modified.

Page 7 line 153 Please, give a information that the gradient method was used during chromatographic analysis, moreover, 3 mm should be changed into 3µm (in case of particles diameter)

Gradient programme has been added to the text and 3 mm has been changed as 3 µm. Thank you.

Page 7 line 161 Please, use the same designation in the description of Table 1 as in Table 1 (eg. „meat” in Table 1 and „beef muscle” in line 161

“Beef muscle” has been changed as “meat”.

Page 8 line 199 „groups by Girard” should be changed as „ groups as reported by Girard”

Thank you. The correction has been done. 

Page 13 line 314 „ranged between nd – 2,75 ng/g” should be changed into „ ranged from 0,5 – 2,75 ng/g, whereas in the case of meatballs cooked at 150°C HAAs were not detected”

Thank you. The correction has been done in the text.

Page 15 line 355 „radical quenchers and free radical scavenging activity are designation of the same phenomenon – antiradical activity

This part has been corrected.

Page 16 line 373 „the chromatographic detector” should be changed as „ type of chromatographic detection”

Thank you. The correction has been done in the text.

Page 16 lines 382&385 The last sentence is not so adequate to the conclusion of this manuscript

The reviewer is right. Therefore, the sentence has been moved to results and discussion section. Thank you.

Table 2 The Authors have given the values of determinated parameters in first line named as Usage rate (0-1) but nothin is known about temperature of cooking (in case of first line), so detailed information could be helpful for the readers (under the table or in the text),

With respect to the values of parameters given in line No 3 named as cooking temperature, did the Authors give the average values for various usage rate (0–1 %), please to explain

Thank you for your comment. Table 2 shows the average values of the results. Therefore, n (number of sample) was added to the Table 2. I think n explains the situation.

The revised manuscript has been resubmitted to your journal. I look forward to your positive response as soon as possible. Is there anything I can do for that, please don’t hesitate to contact me.

Sincerely yours,

Dr. Oz

---

## [Decision Letter · Decision Letter 1]

14 Aug 2019

Inhibitory effects of black cumin on the formation of heterocyclic aromatic amines in meatball

PONE-D-19-18609R1

Dear Dr. Oz,

We are pleased to inform you that your manuscript has been judged scientifically suitable for publication and will be formally accepted for publication once it complies with all outstanding technical requirements.

With kind regards,

Aneta Agnieszka Koronowicz, PhD

Academic Editor

PLOS ONE

Additional Editor Comments (optional):

Reviewers' comments:

Reviewer's Responses to Questions

**Comments to the Author**

1. If the authors have adequately addressed your comments raised in a previous round of review and you feel that this manuscript is now acceptable for publication, you may indicate that here to bypass the “Comments to the Author” section, enter your conflict of interest statement in the “Confidential to Editor” section, and submit your "Accept" recommendation.

Reviewer #1: All comments have been addressed

Reviewer #3: All comments have been addressed

2. Is the manuscript technically sound, and do the data support the conclusions?

Reviewer #1: Yes

Reviewer #3: Yes

3. Has the statistical analysis been performed appropriately and rigorously? 

Reviewer #1: Yes

Reviewer #3: Yes

4. Have the authors made all data underlying the findings in their manuscript fully available?

Reviewer #1: Yes

Reviewer #3: Yes

5. Is the manuscript presented in an intelligible fashion and written in standard English?

Reviewer #1: Yes

Reviewer #3: Yes

6. Review Comments to the Author

Reviewer #1: Dear Editor,

The article is deal with the inhibitory effect of black seed on the formation of heterocyclic amines in meatball. The topic is interest and the article is good and easy to follow. In addition, the author has modified the article according to the reviewers comments. It can be published in your journal with this form.

Sincerely yours,

Reviewer #3: In my opinion the manuscript entitled "Inhibitory effects of black cumin on the formation of heterocyclic aromatic amines in meatball" may be published in PLOS ONE

7. PLOS authors have the option to publish the peer review history of their article (what does this mean?). If published, this will include your full peer review and any attached files.

Reviewer #1: No

Reviewer #3: No

---

## [Editor Report · Acceptance letter]

20 Aug 2019

PONE-D-19-18609R1 

Inhibitory effects of black cumin on the formation of heterocyclic aromatic amines in meatball 

Dear Dr. Oz:

I am pleased to inform you that your manuscript has been deemed suitable for publication in PLOS ONE. Congratulations! Your manuscript is now with our production department. 

With kind regards,

on behalf of

Dr. Aneta Agnieszka Koronowicz 

Academic Editor

PLOS ONE